# Central Nervous System Infections Due to Coccidioidomycosis

**DOI:** 10.3390/jof5030054

**Published:** 2019-06-28

**Authors:** Niki R. Jackson, Janis E. Blair, Neil M. Ampel

**Affiliations:** 1Division of Infectious Diseases, Mayo Clinic in Arizona, Phoenix, AZ 85054, USA; 2University of Arizona College of Medicine, Tucson, AZ 85724, USA

**Keywords:** *Coccidioides*, coccidioidomycosis, central nervous system, meningitis, brain abscess, fungal infections

## Abstract

Coccidioidomycosis is a common infection in the western and southwestern United States as well as parts of Mexico and Central and South America and is due to the soil-dwelling fungi *Coccidioides*. Central nervous system (CNS) infection is an uncommon manifestation that is nearly always fatal if untreated. The presentation is subtle, commonly with headache and decreased mentation. The diagnosis should be considered in patients with these symptoms in association with a positive serum coccidioidal antibody test. The diagnosis can only be established by analysis of cerebrospinal fluid (CSF), which typically demonstrates a lymphocytic pleocytosis, hypoglycorrhachia, elevated protein, and positive CSF coccidioidal antibody. Cultures are infrequently positive but a proprietary coccidioidal antigen test has reasonable sensitivity. Current therapy usually begins with fluconazole at 800 mg daily but other triazole antifungals also have efficacy and are often used if fluconazole fails. Triazole therapy should be lifelong. Intrathecal amphotericin B, the original treatment, is now reserved for those in whom triazoles have failed. There are several distinct complications of CNS coccidioidal infection, the most common of which is hydrocephalus. This is nearly always communicating and requires mechanical shunting in addition to antifungal therapy. Other complications include cerebral vasculitis, brain abscess, and arachnoiditis. Management of these is difficult and not well established.

## 1. Introduction

Coccidioidomycosis is an infection caused by the soil-dwelling fungi *Coccidioides*. It is endemic to areas of the southwestern United States, particularly the San Joaquin Valley of California and the south-central region of Arizona, as well as northern Mexico, and parts of Central and South America. Although uncommon, central nervous system (CNS) infections are among the most devasting manifestations of coccidioidomycosis. In this paper, we review the medical literature and offer our approach regarding the diagnosis and management of various forms of CNS coccidioidomycosis. We spend most of the time on coccidioidal meningitis since it is the most common manifestation of CNS involvement.

## 2. Epidemiology and Risk Factors

CNS involvement by *Coccidioides* involves all age groups, races, and both sexes [1]. However, adolescent and young adult white men appear to be disproportionately affected, especially when CNS disease is the only manifestation of extrathoracic coccidioidal infection. African-American men have higher incidence of extrathoracic dissemination in general and may present with CNS disease in association with coccidioidomycosis at other extrathoracic sites [2]. In the landmark study by Bouza et al. of 31 patients, 27 were male and 19 were non-Hispanic whites. The average age was 41 years [3].

There is no evidence that higher inoculum exposures, such as occur during archeological excavations or other direct work with soils, result in a higher risk of CNS coccidioidomycosis. In addition, it is unclear if immunosuppressive therapy represents an increased risk for the development of CNS coccidioidomycosis over other forms of extrathoracic disseminated disease. In one study of 40 patients receiving immunosuppressive medication, 28 had extrapulmonary disseminated infection and 2 of these had meningitis [4]. In a recent review comparing cases from 2008 [5] to those reported in 1980 [3], no specific underlying illness stood out except for HIV-1 infection, which had not yet been described in 1980. A case series from 1996 to 2007 included 71 patients with CNS coccidioidomycosis referred to a tertiary care center [6]. In this study, disease occurred in all ages and males comprised greater than two-thirds of the study population. African-Americans were at a 6-fold increased risk compared with whites. Only 42% of patients were considered to be immunocompromised and diabetes did not appear to be a significant risk factor.

## 3. Pathogenesis

CNS coccidioidomycosis occurs when coccidioidal spherules or endospores migrate to the meninges or into brain tissue. In an early study of 35 autopsy cases, CNS coccidioidomycosis was noted to have a predilection for the basilar portion of the brain with thickened, hyperemic meninges [7]. In a later study of 32 patients, endarteritis obliterans with inflammatory cells throughout the outer layers of small arteries and arterioles with narrowing of vessel lumina and occluded arteries associated with focal necrosis and inflammatory exudates [8]. In addition, infarcts, principally of the basal ganglia, thalamus, and white matter, were observed. The spinal cords demonstrated thickened meninges with granulomatous inflammation. Nerve roots were frequently entrapped by meningeal exudates and the brain parenchyma was involved in about one-third of the cases. In a review of eight cases over 25 years [9], foci of both granulomatous and suppurative inflammation of the meninges with coccidioidal spherules were seen in all patients. The process often extended into the underlying brain or spinal cord parenchyma with focal granulomata or abscesses. Focal arteritis was seen in half.

## 4. Clinical Presentation

The course of coccidioidal meningitis is insidious and progressive. The patient usually presents with a global headache that may not be severe but is persistent over weeks to months. As the disease progresses, the patient may develop cognitive dysfunction manifesting as mild confusion and emotional lability. This is often first noticed by family members [1]. Gait disturbances, diploplia, disorientation, lethargy, and stupor may be seen, particularly if hydrocephalus has occurred. Untreated, coccidioidal meningitis is nearly always fatal. Buss et al. reported on 53 patients from the era prior to antifungal therapy. Fifty died in a span of six weeks to two years [7]. In a review of 25 documented cases of coccidioidal meningitis from military and veterans’ hospitals when antifungal therapy was not available, 24 died directly from their infection. Fourteen died within eight months of diagnosis and the majority of these were during the first three months [10].

Physical examination demonstrates a paucity of findings. Cognitive slowing may be evident, with slowed speech and inability to understand common tasks. Frank nuchal rigidity is uncommon [2,6] but there may be neck stiffness. Focal neurological findings are rare unless there is a mass lesion. Blurred vision with papilledema seen on ophthalmological examination may occur in cases of hydrocephalus. Evidence of pulmonary or other sites of coccidioidomycosis occurs in about one-half to two-thirds of patients [3,6,10]. When there is a diagnosis of primary pulmonary coccidioidomycosis, symptoms of meningitis tend to occur soon after [11]. In one study, meningitis was seen within three months in 17 of 24 patients with identified pulmonary coccidioidomycosis, and six years was the longest period [3]. Many patients with CNS disease will not have any other clinical manifestation of coccidioidomycosis and this should not dissuade from the diagnosis.

## 5. Diagnosis

A positive serum coccidioidal antibody test is nearly universal in patients with coccidioidal meningitis [3]. Therefore, the presentation of a persistent headache, cognitive decline, personality changes, alterations in vision, gait disturbance, or focal neurological deficits in association with a positive serological test should lead to considering the diagnosis of coccidioidal meningitis. The only definitive test is analysis of cerebrospinal fluid (CSF). This must be performed to establish the diagnosis.

Headache is also a predominant symptom of primary pulmonary coccidioidomycosis without meningitis [12] and it can be difficult to distinguish the two entities. If there is any doubt, a CSF analysis should be done. The CSF abnormalities persist for weeks to months after triazole antifungal therapy is initiated [13]. Therefore, if the headache is persistent in a patient with primary pulmonary coccidioidomycosis, it is appropriate to perform a CSF analysis even after antifungal therapy has been initiated.

The value of performing a diagnostic lumbar puncture in the absence of headache or other neurological symptoms in patients with a diagnosis of coccidioidomycosis was addressed by Thompson et al. in a retrospective chart-review study [14]. Of the 14 patients with a final diagnosis of coccidioidal meningitis based on CSF results from lumbar puncture, all had a headache. Among the 44 patients in whom this diagnosis was excluded based on CSF analysis, only 18 had a headache. The absence of a headache or another CNS disease sign or symptom had high negative predictive values, suggesting that in the absence of such symptoms, a CSF analysis is not useful.

The CSF findings of coccidioidal meningitis include a leukocyte pleocytosis of between 20 and 2000 cells/µL that is usually lymphocytic but is sometimes polymorphonuclear. In one study, the average number of white cells was 263/µL [3]. A particularly suggestive finding is CSF eosinophilia [15,16]. Hypoglycorrhachia is frequent and often below 40 mg/dL [6]. Protein concentrations are variable but can be strikingly elevated if hydrocephalus is present. Opening pressure should always be measured in suspected cases and an elevated pressure suggests hydrocephalus.

In addition to the routine CSF studies for infectious meningitis, there are a series of special studies that should be considered when coccidioidal meningitis is suspected. The first is the complement-fixation (CF) coccidioidal antibody test, which measures IgG antibody. Over 95% of CSF samples are positive for CF antibody in cases of coccidioidal meningitis [17,18]. This assay should be performed on undiluted CSF and may be done either by traditional complement-fixing methods or by immunodiffusion. Concentrating the CSF or performing the assay by EIA (IgG EIA) may result in a false positive [18]. Unlike in the serum, the titer of the CF antibody response in the CSF does not have prognostic implications [17]. Tube precipitin (TP) IgM antibody may be found in the CSF in a minority of patients and need not be obtained since it does not add to the diagnosis [17,18]. In rare instances, a positive CSF CF antibody test may be seen when there is juxtadural infection, such as vertebral osteomyelitis or paravertebral abscess, without meningitis [1]. In such cases, the CSF is usually acellular, hypoglycorrhachia is lacking, and imaging studies demonstrate the abnormality.

Fungal culture is positive in fewer than one-third of cases. Even less common is the finding of fungal elements by direct microscopy [1,2,3,11,19]. In either case, such a finding is diagnostic.

Several nonculture techniques to detect *Coccidioides* in clinical samples are now available. One is measurement of (1,3)-β-glucan in serum and other body fluids. This dextrose polymer is present in the cell wall of most fungi, including *Coccidioides*. Stevens et al. [20] found that the test had a sensitivity of 96% with a specificity of 82% at a cut-off of 31 pg/mL using CSF samples from a cohort of 21 patients with coccidioidal meningitis. However, both the sensitivity [21] and specificity [22] of this assay as a general test for the diagnosis of coccidioidomycosis have been challenged. In our view, it currently has a limited role in the diagnosis of CNS coccidioidomycosis.

Real-time polymerase chain reaction (RT-PCR) is a more specific approach. Binnicker and colleagues designed primers that target the ITS2 region of *Coccidioides* and reported 100% sensitivity of RT-PCR compared to culture using a variety of clinical samples with results available in hours [23]. They subsequently reported its successful use in two cases of coccidioidal meningitis [24]. However, in a retrospective review of five CSF samples from two patients with coccidioidal meningitis [25], the assay was negative in both while the culture was positive in one. These results suggest that the assay may lack the requisite sensitivity for diagnosing coccidioidal meningitis.

An antibody enzyme immunoassay directed against the galactomannan antigen expressed by *Coccidioides* has been developed for both urine and serum [26] as a proprietary test. This has recently been used to test CSF samples from patients with coccidioidal meningitis. In one retrospective review of 36 patients with presumed coccidioidal meningitis compared to 88 patients with meningitis due to other etiologies, a sensitivity of 93% and a specificity of 100% were reported [27]. In another retrospective review of seven cases of coccidioidal meningitis, five of whom were immunocompromised, measuring CSF coccidioidal antigen was useful both for diagnosis and management [28]. These reports suggest that determining CSF antigen is helpful in diagnosing and monitoring the progress of coccidioidal meningitis.

The site where the CSF is obtained may affect results. Samples obtained by lumbar or cisternal puncture provide the most accurate profile and are preferred. Samples obtained from ventricular fluid, as occasionally occurs, may be misleadingly benign [29].

Neuroimaging should always be obtained in cases of suspected CNS coccidioidomycosis but should not be used as a substitute for obtaining a CSF sample. CT scans with contrast and MRI with gadolinium both provide useful clinical data. However, MRI detects abnormalities more frequently than does CT [30] and has the ability to detect meningeal enhancement, making it the preferred modality. Results of enhanced MRI typically reflect the inflammation of the basilar cisterns that is described in pathological studies by demonstrating enhancement of this region as well as of the sylvian fissures and the pericallosal regions. Deep infarcts are also commonly observed [31,32]. A critical finding for management is whether hydrocephalus is present. The finding of hydrocephalus presages a poor outcome [30].

Two recent reports have demonstrated the importance of imaging to detect spinal as well as intracranial abnormalities in coccidioidal meningitis. Lammering et al. [33] reviewed MRI imaging in 23 cases of coccidioidal meningitis from 1998 to 2011. They found leptomeningeal enhancement of the basilar cisterns in 91% and hydrocephalus in 78%. Irregularities in cranial vessels were observed in 30% when MRA was performed. Concomitant spinal abnormalities were seen in 86% of patients on spinal imaging, including 74% showing leptomeningeal enhancement, 37% with nerve root clumping and thickening, 16% with intramedullary spinal cord T2 signal hyperintensity, and 16% with intradural masses. In those with myelography, complete CSF blocks were seen frequently and some developed over time. Crete et al. reviewed their six-year experience with spinal imaging among 41 patients [34]. Overall, 73% had intrathecal involvement. Leptomeningeal enhancement was most often diffuse, involving the entire cord. Adhesive arachnoiditis, manifested most often by nerve root clumping, was observed most frequently in the lumbar spine. Cord edema and syrinx were observed equally in the cervical and thoracic cord. These studies suggest that in addition to intracranial imaging, spinal imaging should be considered in all patients with coccidioidal meningitis.

## 6. Treatment

A synopsis of the various treatment approaches to coccidioidomycosis are displayed in Table 1. While intravenous amphotericin B deoxycholate was the first successful therapy for coccidioidomycosis, it was soon recognized that it was ineffective in treating CNS disease. The use of intrathecal amphotericin B deoxycholate for coccidioidal meningitis was pioneered by Winn [35] and extended by Einstein et al. [36]. Labadie and Hamilton were able to increase the amount of amphotericin B deoxycholate given per injection by slowly increasing the dose and combining each injection with hydrocortisone [37]. Intrathecal therapy can be administered by several routes, including directly into the cisternal space, into the lumbar space using a hyperbaric solution and Trendelenburg positioning [38], through a lateral cervical approach [1], and into a ventricular reservoir [39]. Except for the latter, which is the least effective, all require considerable expertise [40] and can have significant complications, including bleeding, secondary bacterial infection, and arachnoiditis [11]. Hence, these injections should only be attempted by someone experienced in these procedures. Berry et al. [41] have recently reported on the successful use of an intracisternal catheter attached to a programable pump that allowed continuous infusion of amphotericin B deoxycholate. Unfortunately, relapses once intrathecal therapy is discontinued are frequent and ultimate cure appears to be only about 30% [1]. While newer lipid formulations of amphotericin B are now available, these have not generally been used for intrathecal injection. However, in a recent report, 1.5 mg of liposomal amphotericin B was injected into a ventricular port and given repeatedly over a six-week period. The patient did develop ventriculitis, but it was unclear if this was due to the amphotericin B injections or another process [42].

A revolution in management occurred with the study of oral triazole antifungals for coccidioidal meningitis in the 1990s. Fluconazole was initially studied because of its substantial CSF levels after oral administration [43]. An uncontrolled trial of fluconazole at 400 mg daily found that 37 of 47 evaluable patients responded to treatment [13], a result echoed by an earlier study [44]. Itraconazole, which does not reach significant concentrations in the CSF, has also been shown to be effective [45].

Current clinical practice is to begin therapy with fluconazole 800 mg daily. After months of treatment with evidence of clinical improvement, some clinicians would reduce that dose to 400 mg daily, but others would continue at 800 mg daily. In either case, therapy would then be maintained lifelong given a relapse rate of nearly 80% once therapy is discontinued [46]. In patients doing well, there is no imperative for repeating CSF or neuroimaging. However, it is important to clinically monitor the patient as well as follow the serum coccidioidal serologies over time. A stable clinical presentation with a downward trend in serum coccidioidal titers is supportive of control of infection. Neuroimaging and repeat analysis of CSF should be immediately pursued if the patient develops clinical deterioration or any neurological symptoms.

There are several approaches in managing patients that fail fluconazole therapy. Some clinicians increase the fluconazole dose up to 1200 mg daily [47]. This may be effective but increases the risk of toxicity, including alopecia, dry skin and lips, and hepatitis. More common is to change to another triazole, most frequently either voriconazole or posaconazole. This is based on results from case reports [48,49,50,51] as well as recent data that in vitro susceptibility of *Coccidioides* is much less to fluconazole than to other triazole antifungals [52]. The use of voriconazole and posaconazole in these situations must be tempered by issues of increased cost, adverse events, drug interactions, and the need for therapeutic drug-level monitoring [53]. There is now experience with isavuconazole for coccidioidal meningitis. In nine patients who were either fluconazole failures or had drug toxicity from voriconazole, five patients were judged as successes and the other four were considered stable [54]. Based on this, isavuconazole is another potential alternative therapy.

An intriguing approach is to use an intravenous lipid-formulation amphotericin B. This is based on an experimental rabbit model of coccidioidal meningitis pioneered by Clemons and colleagues where either liposomal amphotericin B or amphotericin B lipid complex appeared as good as or better than fluconazole [55,56] and equivalent to each other [57]. In clinical practice, Antony et al. have reported a case [58] where intravenous liposomal amphotericin B at 3 mg/kg daily was combined with voriconazole 200 mg daily in a patient with presumed meningitis and brain lesions who had failed prior therapy. After improvement and 16 months of treatment, the liposomal amphotericin B was discontinued and the patient was kept on voriconazole alone. Kuberski, in an unpublished report [59], details a possible cure with an initial four-week course of intravenous liposomal amphotericin B at 5–7 mg/kg per dose followed by oral fluconazole that was subsequently discontinued. While we cannot advocate this sequential approach for coccidioidal meningitis, it may be reasonable in selected relapsed cases to use a lipid preparation of amphotericin B in combination with a triazole antifungal. The latter would continue indefinitely after the amphotericin B is discontinued.

In cases that do not respond to more conservative therapy, intrathecal amphotericin B therapy can be started alone or with an oral triazole. This is the currently the main indication of intrathecal therapy [40]. As noted above, it should only be administered by clinicians experienced with its administration. The length of intrathecal therapy is unclear. It has been recommended to continue until the coccidioidal CF titer in the CSF is no longer detectable. This may be months to years [1].

Adjunctive corticosteroids have been used empirically in coccidioidal meningitis in cases complicated by vasculitis and stroke. Recently, a retrospective review from several medical centers examined the outcome of 18 patients who had cerebrovascular events associated with coccidioidal meningitis [60]. Fifteen received some course of corticosteroids subsequent to the event compared to three who did not. Patients who received corticosteroids for secondary prevention were significantly less likely to develop additional cerebrovascular events compared with those who did not. Unfortunately, this study did not answer the larger question of whether all patients with coccidioidal meningitis should receive corticosteroids. At this time, such therapy is not routinely given.

While interferon-gamma has been used as an adjunctive therapy in case reports of severe and unresponsive coccidioidomycosis [61,62], there are no supportive data available to make a recommendation either for or against this for CNS disease. If other therapeutic modalities have failed, it is not unreasonable to consider its use.

One question is if and when immunosuppressive medications can be used in patients with CNS coccidioidomycosis who require these agents to treat other medical conditions, such as a rheumatological disease or inflammatory bowel disease. There are no clear guidelines. We favor restarting these once the patient appears to be clinically improved on antifungal therapy.

## 7. Complications

The most frequent complications of CNS coccidioidomycosis are increased intracranial pressure with hydrocephalus, brain abscess, vasculitis, spinal arachnoiditis, and syrinx formation [11,47,63]. Increased intracranial pressure with hydrocephalus is the most common of these. It may occur as part of the initial presentation or after antifungal therapy has been initiated. Typically, the patient presents with increasing and persistent headache with or without decreased mental acuity. In such cases, neuroimaging with subsequent lumbar puncture and measurement of opening pressure should be performed as soon as possible [47]. Hydrocephalus associated with coccidioidal meningitis does not usually resolve with antifungal therapy alone or when combined with repeated removal of CSF by lumbar puncture and almost always requires anatomic shunting. It is usually of the communicating variety where CSF flow from the ventricles is blocked due to basilar inflammation [64,65]. Consultation with neurosurgery should occur early to define the best approach to correct the abnormality. While placement of a ventricular drainage shunt is corrective, shunt failure occurs in at least half the patients and requires surgical revision, often multiple times. Mortality in patients who develop hydrocephalus approaches 10% even with shunting [66]. The timing of CSF shunt placement is controversial. Many neurosurgeons prefer to wait until there is clearance of inflammation, but this is usually not feasible. We favor placement as soon as the diagnosis of hydrocephalus is established.

Symptomatic brain abscesses are an uncommon manifestation of CNS coccidioidomycosis. The presentation varies widely among reported cases, but some patterns have emerged. The majority of reported cases have occurred in patients with altered immunity, including diabetes mellitus, HIV-1 infection, solid organ transplantation, malignancy, and treatment with immunosuppressive medications. There is a male predilection [6,67,68,69,70,71]. Presenting symptoms vary and encompass all the symptoms that suggest a mass CNS lesion, including ataxia, sensory deficits, motor deficits, cranial nerve palsies, altered mental status, and headache. Imaging generally reveals single or multiple hypodense lesions with ring enhancement. The mechanism by which *Coccidioides* causes brain abscess is unclear but appears to occur in patients with [6] and without evidence of meningitis [67,69,70]. Therefore, it is postulated that brain abscess can occur as a consequence of direct spread from a meningeal lesion or may occur as a result of hematogenous spread directly into brain parenchyma. While there is no consensus on management and mortality appears to be high, biopsy to confirm the diagnosis followed by prolonged if not lifelong antifungal therapy is most appropriate. In addition, coccidioidomycosis may cause intramedullary spinal masses, often presenting with paresis or paralysis [72,73,74,75]. Treatment has generally included antifungal therapy, corticosteroids, and ventricular shunting when hydrocephalus is present [72,73,74]. 

Vasculitis was recognized early as a pathological event in CNS coccidioidomycosis [8] but was not well described as a clinical entity until later [76,77]. Williams et al. reported 10 cases in 1992. In their series, patients often presented with abrupt declines in mental status associated with symptoms of vascular insufficiency, such as aphasia, hemianopsia, and hemiparesis. CSF findings were typical of uncomplicated coccidioidal meningitis and were not predictive of this complication. Initial imaging was unremarkable but later showed focal parenchymal involvement with infarction [78]. The precise pathophysiological events responsible for vasculitis in coccidioidal meningitis have not been elucidated but an excess of metalloproteinase-9 produced by inflammatory cells was described in an animal model [79]. The most appropriate management is unclear [80]. In addition to antifungal therapy, many clinicians will add corticosteroids to prevent subsequent events [60]. A distinct complication is subarachnoid hemorrhage due to the development of arterial aneurysms [81]. These are often multiple and may require neurosurgical intervention [82,83].

Spinal arachnoiditis occurs in up to 10% of patients with coccidioidal meningitis [5]. In severe cases, this may result in neurological symptoms, including lumbar pain, neurogenic bladder and paresis [47], as well as lower extremity weakness and ataxia [84]. Adhesive arachnoiditis may result in paresis from blockage of CSF flow [33], precluding the use of intrathecal amphotericin B.

A syrinx is a fluid-filled cavity of the spinal cord or brain stem. It occurs infrequently in coccidioidal meningitis, usually when there is obstruction to CSF flow. Due to its mass effect, it can result in paresis. A syrinx can be distinguished from cord edema by the degree of hypointensity on T1 MRI imaging, where the syrinx is closer to CSF intensity [34]. Syrinxes tend to resolve with resolution of the CSF obstruction, so shunting or neurosurgical relief over the involved area are the treatments of choice [85].

## 8. Summary

Although CNS involvement is an uncommon complication of coccidioidomycosis, it is devastating when it occurs. Diagnosis always requires analysis of CSF, which usually demonstrates hypoglycorrhachia and elevated protein with a positive CSF coccidioidal IgG response. Initial therapy is usually fluconazole from 400 to 800 mg daily continued indefinitely. The most common complication is communicating hydrocephalus, which almost always will require CSF shunting. Other complications include abscess, vasculitis, and arachnoiditis. Management usually requires considerable expertise and this should always be sought when the diagnosis is established.

## Figures and Tables

**Table 1 jof-05-00054-t001:** Summary of medications used for the treatment of central nervous system (CNS) coccidioidomycosis.

Antifungal Agent	Typical Dose/Frequency	Notes	Comments
**First-line treatment**			
Fluconazole	400–800 mg orally daily	• absorbed well orally	• dosing to 1200 mg/daily can be considered if patient has a clinically inadequate response
**Alternatives to first-line agent**			
Itraconazole	200 mg orally BID–TID	requires acid for absorption acidic food/drink may improve absorption requires drug-level monitoring of itraconazole and hydroxyitraconazole combined total levels 3.0–6.0 µg/mL recommended	numerous drug–drug interactions black box warning for patients with heart failure poor cerebrospinal fluid (CSF) penetration, but affinity for P-glycoprotein results in active efflux
**For failures of initial therapy**			
Voriconazole	200–400 mg orally BID	good oral bioavailability less CNS penetration compared with fluconazole target serum levels 3–6 µg/mL	numerous drug–drug interactions expensive cutaneous phototoxicity reversible visual changes
Posaconazole	300 mg daily as delayed-release (DR) preparation	poor CNS penetration target of serum levels 3–6 µg/mL	high protein binding but large volume of distribution serum levels > 5 may increase likelihood of toxicity inhibits renin–angiotensin–aldosterone axis in some patients
**Rescue therapy for azole-intolerant infection**			
IT amphotericin B deoxycholate (AMBd)	escalated doses as tolerated: initiate dosing as 0.1 mg 3 times weekly gradual increase of dose by 0.1 mg as tolerated weekly dose escalation by >0.1 mg/dose may not be tolerated	treatment of choice prior to established role of azoles currently used for azole failure	nausea, vomiting, and headache are common premedication with acetaminophen, diphenhydramine, or ondansetron orally may be helpful concurrent IT corticosteroid reduces risk of arachnoiditis neurotoxic effects include ataxia, erectile dysfunction, hearing loss, ophthalmoplegia, neurogenic bladder, and paraplegia
**Less well established adjunctive therapy**			
Dexamethasone	20 mg daily × 7 days, then lower dose by 4 mg every other day		• may be useful in cases of vasculitis

IT, intrathecal; mg, milligram.

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
