# Peer review of "Central Nervous System Infections Due to Coccidioidomycosis"

_jof, 2019, doi:10.3390/jof5030054_

Round 1
Reviewer 1 Report
Well written review of the topic. My only suggestion is a final summary paragraph.
Author Response
A summary paragraph has been added to the manuscript.
Reviewer 2 Report
A very well written review of an important topic.
The review includes current/state-of-the-art data on this topic and provides detailed guidance for healthcare professionals caring for patients with this challenging infection.
Consider adding the following data to the paper-
-The role of trending CF titers in serum among these patients.
-Timing of placement of VP shunt among those with hydrocephalus; should a period of antifungal therapy be given prior to placement, etc.
-Use of gamma interferon or other treatment options for salvage therapy
-Timing for use of immunosuppressive agents for those with co-existing conditions (autoimmune diseases needing TNF blockers, etc).
Excellent review overall.
Author Response
We have added statements regarding the use of trending serum coccidioidal serology, the controversy regarding when a CSF shunt should be placed, the use of immunosuppressives, and the use of interferon-gamma.